# The location of an internal focus of attention differentially affects motor performance

Andrew J. Strick[1]*, Logan T. Markwell[1], Hubert Makaruk[2], Jared M. Porter[1]

1 Department of Kinesiology, Recreation and Sport Studies, University of Tennessee Knoxville, Knoxville, Tennessee, United States of America, 2 Department of Physical Education, Józef Piłsudski University, Warsaw, Poland

* astrick@vols.utk.edu

**Data Availability Statement:** All relevant data are within the manuscript and its Supporting Information Files.

**Funding:** No funding was attained or acquired for this study.

## Abstract

Prior research has questioned the appropriateness of internal focus instructions or the juxtaposition to external and control conditions. Moreover, there has been a lack of research conducted to test a variety of internal conditions on motor performance. The purpose of the present study was to address those critiques and add to the attentional focus literature by varying the location of an internal focus while performing a standing long jump. Participants performed a standing long jump during five separate conditions (internal focus: toes, knees, hips, arms; and control). The analysis revealed that all internal focus conditions performed worse than the control condition. Furthermore, the only difference between internal conditions was found between arms and knees, where the knee condition resulted in a significantly shorter jump distance relative to the arms. Regardless of the location specified, all internal focus conditions led to detriments in jump performance when compared to the control condition. These findings add to a large body of work demonstrating the importance of instructional content on motor performance.

## Introduction

Since the inception of focus of attention research, numerous studies have found that directing attentional resources externally towards the movement effect improves motor performance and learning compared to an internal focus or control conditions (for a review, see [1]). For instance, when hitting a golf ball, an external focus of attention (EFOA) would be a focus on the movement of the golf club during task execution. In contrast, adopting an internal focus of attention (IFOA) for the same task would direct attention towards body movements of the arms or rotation of the torso when swinging the golf club. The motor performance and learning differences from these two separate ways of directing attentional resources have been explained by the constrained action hypothesis [2]. The constrained action hypothesis proposes the motor control system operates more autonomously when attention is directed externally. Whereas the motor control system is disrupted when attention is focused internally.

The predictions made by the constrained action hypothesis have been rigorously tested across numerous experimental paradigms [1]. Disruption of the motor control system when focusing attentional resources internally has been observed in a variety of studies pairing

**Competing interests:** The authors have declared
that no competing interests exist.

outcome results with measures of electromyography (EMG) [3–6], movement production [7,8], and central nervous system activity [7,9,10]. Studies investigating attentional resources effect on maximal force production using EMG have found disruption of automatic control processes via greater muscle activation paired with decreased maximal force output [5]. Furthermore, other EMG based studies have shown greater co-contraction of agonist and antagonist musculature when focusing internally paired with decreased motor performance in the vertical jump [6] and dart throw [3,4]. Additionally, Kal et al. [8] tested predictions of the constrained action hypothesis by measuring movement automatization in a cyclic single leg extension-flexion task. Results from their study found shorter movement times and more regular movement execution when focusing externally compared to focusing internally, which suggests that an internal focus disrupted more regular automatic movement patterns when compared to an external focus. Furthermore, findings from Huang et al. [7] showed that an EFOA elicited greater postural regularity and greater brain activation in the right temporo-parietal junction associated with force matching compared to an IFOA when performing a stabilometer balancing task. Results from these studies show greater movement regularity with an EFOA compared to an IFOA. Other attentional focus research measuring central nervous system activity found greater inhibition in the primary motor cortex (M1) and longer durations in a timed task till failure for an EFOA compared to an IFOA [9]. Their findings suggested this inhibitory motor control was disrupted while focusing internally, contributing to greater movement inefficiency compared to focusing externally. In addition, other work investigating attentional focus on neural pathways found that focusing externally during a foot pedal pressing task led to greater activation of faster neural pathways, whereas focusing internally resulted in slower motor pathways [10]. In sum, the evidence provided from a multitude of different measures supports the predictions made from the constrained action hypothesis [2]. An IFOA seems to disrupt the automatic control processes of the motor system, whereas an EFOA facilitates automatic control processes.

One of the ways practitioners and researchers implement focus of attention is through the delivery of verbal instructions. In furthering this area of research, McNevin et al. [11] used a dynamic balancing task to compare the distance of near versus far EFOA instructions. Instructions in the near EFOA (i.e., proximal) condition directed attentional focus at markers in close proximity to the mover's feet, whereas the far-outside and far-inside EFOA (i.e., distal) conditions were instructed to focus attentional resources at markers placed distally from the mover: near the outside edges, and at the center of the balance platform. McNevin et al. [11] found motor learning benefits for both of the far EFOA conditions compared to the near EFOA condition. Their results advanced focus of attention research by providing initial evidence for greater motor learning benefits for instructions that direct attention externally more distally compared to proximally.

This distance effect was further expanded to include motor performance in the standing long jump (SLJ) by Porter et al., [12,13]. In their initial study [12], for each SLJ attempt the IFOA group was instructed to "...focus your attention on extending your knees as rapidly as possible. In contrast, the EFOA group was instructed to "...focus your attention on jumping as far past the start line as possible". Results from their study showed evidence for greater SLJ distance when focusing externally compared to internally. Porter et al. [13] extended their previous work using instructions to test distal versus proximal external focus on SLJ performance. Findings from this study confirmed earlier work from McNevin et al., [11] as the data showed that horizontal jump distance was systematically increased as attention was directed at spatially greater distances from the mover. Since the aforementioned findings, much research using the SLJ has been conducted exploring the distance effect on motor learning and performance with regard to the external far condition [14–18].

Other research has investigated adjustments made to IFOA conditions. Recently, Pelleck and Passmore [19] investigated two separate internal foci compared to an EFOA when putting a golf ball. Participants were instructed to focus on the hands and elbows (i.e., proximal internal), focus on the weight of their feet (i.e., distal internal), or focus on the target (i.e., external). They found the novice's golf putting accuracy was worse for the distal internal focus at the 5-meter distance compared to the other conditions. However, the authors noted that their results can be interpreted as a task relevant (proximal) and task-irrelevant (distal) attentional focus since the distal focus may not be as relevant to the golf putting task, and thus, may not be an equivalent comparison. Separate from task relevancy, Becker and Smith [20] compared a broad IFOA (e.g., focus on using the legs) and a narrow IFOA (e.g., focus specifically on extending the knees) relative to using an EFOA (e.g., jump past the start line) when performing the SLJ. Not only did their findings show that an EFOA led to the greatest performance, which is in line with previous work, but they also revealed that there were no differences between the broad or narrow internal focus conditions. While previous research has examined internal attentional focus in relation to performance, these studies have focused on comparing attention directed to a task-relevant or task-irrelevant cue [19] or attention allocation strategies categorized as board or narrow [20]. Taken together, adjustments investigating changes including altering focus of attention distance [19] or changing focus of attention breadth [20], but very little research has been conducted investigating different IFOA locations. Using the task of long-distance baseball throwing, Oki et al. [21] provided instructions comparing two internal focus conditions, quick wrist flexion and twisting of the torso, relative to external focusing instructions on the ball trajectory in skilled baseball players. Their results showed that focusing on wrist flexion resulted in significantly shorter throwing distance compared to the other two conditions. Interestingly, there was a marginal increase in throwing distance for the EFOA condition compared to the torso internal condition, but it did not meet statistical significance. Separately, Coker [22] had children perform the SLJ to investigate multiple attentional focus instructional cues on motor performance. Findings from this study showed that an external focus led to longer SLJ performances compared to internally focusing on the arms swinging as rapidly as possible and knees extending as rapidly as possible. Coker [22] concluded that their results do not match previous work and could be indicative that children do not have the same level of constrained action as adult movers when performing a horizontal jump.

A variety of critiques have been put forth against research that claims there are detrimental effects on performance when using an internal attentional focus [23]. Additionally, various researchers have questioned the validity of some instructions by their appropriateness or relevant comparison for a given motor skill [23,24]. Others have proposed the value of using internal oriented cues to feel the movements [25]. Moreover, others have suggested the possibility that internal instructions may benefit novices in learning the fundamental dynamics of the movement technique [26]. Given these considerations, the purpose of the present study was to investigate how changes to the location of an internal focus affects standing long jump performance. This differs from previous investigations as this study specifically examined how various internal foci impact motor performance. Based on previous attentional focus work using the SLJ task [12–14,16,20,22,27–30], it was predicted that worse performances would be observed in all internal focus conditions compared to a control condition. Additionally, it was predicted that there would be no significant differences between the IFOA conditions. Such findings would indicate that an internal focus of attention, regardless of its location, is detrimental to the execution of the SLJ. Establishing detrimental equality across a breadth of internal foci is important from both a practical and theoretical perspective. Previous studies examining the focus of attention effect have consistently demonstrated that directing attention externally results in greater SLJ jump distance relative to directing attention internally

[12,13,17]. Furthermore, a recent meta-analysis validated the robustness of the effect across multiple jumping actions [31]. As a result, we did not include an EFOA condition in the present study. Rather, we sought to specifically investigate how jump distance was impacted by various forms of an IFOA relative to jumps completed in a control condition. To be precise, we chose task-relevant instructions that referenced gross body actions to prevent the confound of comparing task relevancy and irrelevancy, on an IFOA location when performing the SLJ.

## Method

### Participants

Consistent with the population and sample size used in similar previous research [31], a total of twenty-nine ($M$ = 21.24, $SD$ = 1.43) male undergraduate college aged students volunteered to participate in the experiment. Only males were included in this study to eliminate sex based differences. None of the participants received previous training in the SLJ. Additionally, none of the volunteers were current or former collegiate or professional athletes. Participants were naïve to the purpose of the study in that they were not informed of the focus of attention predictions on performance. All participants read and signed a written informed consent form before participating in the experiment. A university's Institutional Review Board approved all forms and methods prior to the recruitment of participants.

### Apparatus and task

All data collection took place in a climate-controlled research laboratory. The standing long jump task used in the present study was identical to the jumping task used in previous focus of attention research [4,5,9,10]. All participants were instructed that the goal was to perform the jumping task to the best of their ability. A large black rubber mat of 4.57 meters in length and 0.61 meters in width (Power Systems, Knoxville, TN) with 0.5 inches (1.27cm) between measurement lines running the length of the mat was used for the experiment. Each measurement line on the mat measured 0.125 inches (0.318 cm) in width. After each standing long jump trial, jump distance was measured (in inches) from the start line to the heel of the foot nearest to the start line. This measurement was recorded and later converted into centimeters (cm) for analysis. Before each jump, participants were asked to hold their position after landing so the researcher could measure their performance. A within-participant design was used, where every participant performed two trials under each condition. Condition order was randomized across participants in an attempt to minimize order effects, Table 1.

### Procedures

Upon entering the laboratory, each participant was asked to read and sign an informed consent. Consistent with previous research [12,13,17,18], participants then performed a 5-minute

**Table 1. Count and percentage of first and last condition tested.**

|  | Count of Condition Order | | Percentage of Condition Order | |
| --- | --- | --- | --- | --- |
|  | **First** | **Last** | **First** | **Last** |
| Toe | 5 | 12 | 17% | 41% |
| Hip | 7 | 7 | 24% | 24% |
| Knee | 8 | 2 | 28% | 7% |
| Arms | 2 | 2 | 7% | 7% |
| Control | 7 | 6 | 24% | 21% |

brisk walking warm-up. Upon completion of the warm-up, participants performed a total of 10 standing long jumps. For this study, there were a total of five conditions. Specifically, there were four different task-relevant internal focus conditions with participants being instructed to direct their attention towards specific parts of the body (i.e., toes, knees, hips, arms) prior to the jump. Attentional instructions in this study followed similar cues given in previous work [12,17,22]. The specific instructions for each internal condition were as follows:

Toe: I want you to focus on your toes. When jumping as far as possible focus on pushing your toes down into the floor.

Knee: I want you to focus on your knees. When jumping as far as possible focus on extending your knees as fast as you can.

Hips: I want you to focus on your hips. When jumping as far as possible focus on pushing forward with your hips as fast as you can.

Arms: I want you to focus on your arms. When jumping as far as possible focus on forcefully swinging your arms forward during the jump.

Participants also completed two trials in a control condition which asked them to "perform the jumping task to the best of your ability." For all conditions, before each jump, the researcher provided verbal instructions to participants. Verbal instructions in all four internal focus conditions were used to direct attentional focus. No attention directing cues were provided prior to trials completed in the control condition. Participants performed two consecutive jumps in each of the five conditions for a total of 10 jumps. Each jump was separated by a one-minute rest before moving onto the next attempt. Participants were not informed of their jump distance following any trial and no verbal encouragement was provided to the participants. Following the completion of the study, participants were thanked for their involvement and then they left the testing facility.

## Statistical analysis

The statistical analysis was performed using the statistical package for social sciences (SPSS) version 27. A one-way repeated measures analysis of variance (ANOVA) was performed to evaluate the potential differences in standing long jump distances among the five experimental conditions. A Shapiro-Wilk test was used to test the normality assumption and a Mauchly's test of sphericity was used to test the assumption of sphericity. A post-hoc power analysis using G*Power version 3.1 was used to measure the power of the study [32]. Post-hoc simple effects paired t-tests were used to differentiate between the experimental conditions. Both trials within each condition were averaged for the analyses. Effect sizes for the repeated measures ANOVA were calculated using partial Eta-squared analysis and Cohen's *d* [33] were calculated for post-hoc pairwise comparisons.

## Results

Mauchly's test of sphericity indicated no violation of unequal variances ($p > .05$), so sphericity was assumed. A Shapiro-Wilk test found that only the hip condition ($p = .048$) violated the normality assumption. There were two outliers in the hip condition; however, removal of the outliers led to no statistical differences in our analysis. Therefore, all the data collected was kept for analysis, see Table 2 for descriptives. The ANOVA indicated there was a statistically significant main effect for condition, $F(4, 28) = 7.78$, $p < .001$, $\eta_p^2 = .22$. For the effect size, a sample of 29 participants and an alpha level of 0.05, the post-hoc power analysis showed the study was sufficiently powered with a 1-β of 0.99. Post-hoc simple effects analyses revealed there was a statistically significant difference from which greater average jump distance was observed in the control condition compared to all other conditions: toe, $t(28) = 3.78$, $p < .001$,

**Table 2. Descriptive statistics for standing long jump conditions.**

| Condition | M | 95% CI | SD |
|---|---|---|---|
| Toe | 195.10 | 187.12–203.07 | 20.96 |
| Hip | 192.76 | 184.35–201.17 | 22.11 |
| Knee | 191.86 | 183.20–200.51 | 22.75 |
| Arm | 198.51 | 190.05–206.98 | 22.25 |
| Control | 206.24 | 196.78–215.71 | 24.89 |

M, mean; CI, confidence interval; SD, standard deviation.

Error bars reflect standard error. $^*p < .05$. Jump distance in the Control condition was significantly further than the other four conditions. Additionally, the Arm condition had a greater jump distance than the Knee condition. No other differences were observed.

$d = .70$, hip, $t(28) = 3.77$, $p < .001$, $d = .70$, knee, $t(28) = 4.17$, $p < .001$, $d = .78$, and arm, $t(28) = 2.78$, $p = .010$, $d = .52$ (Fig 1). Furthermore, post-hoc analysis revealed that there was a statistically significant difference where a greater average jump distance was observed in the arm ($M = 198.51$, $SE = 4.13$) compared to the knee ($M = 191.86$, $SE = 4.22$) conditions, $t(28) = 2.52$, $p = .018$, $d = .47$. No other differences were observed.

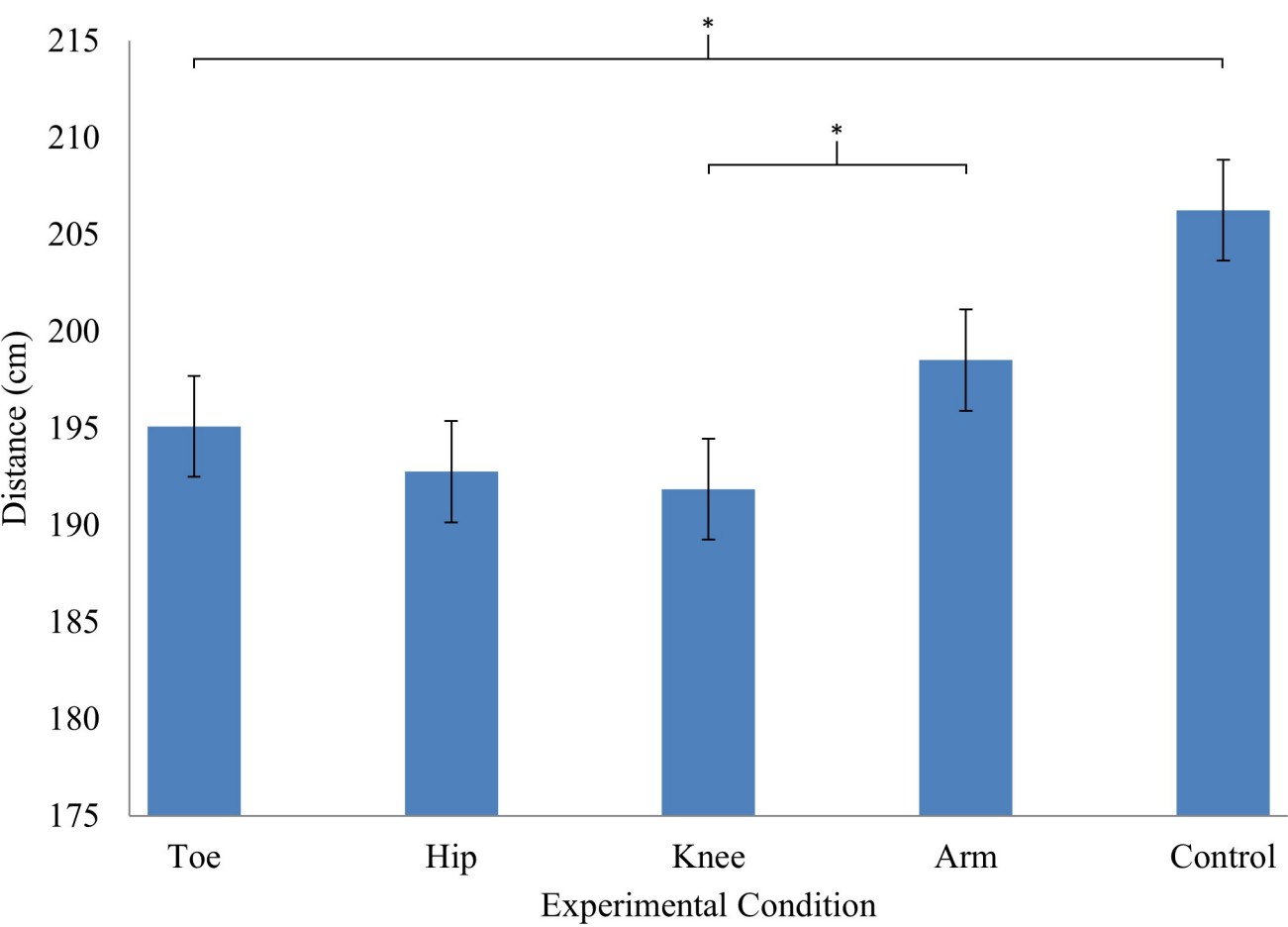

**Fig 1. Standing long jump distance measures.**

## Discussion

Much research has been conducted in the past two decades investigating how altering a mover's focus of attention impacts motor performance and learning [2]. An overwhelming majority of the studies conducted on this topic have found that an EFOA leads to improved motor skill performance and learning compared to an IFOA (see [1,31] for recent meta-analyses and reviews). In addition to the beneficial effects of an external attentional focus, this literature provides evidence that an internal focus of attention has detrimental effects on performance compared to both an external condition and a control condition in which no-attention directing instructions are provided [1]. While these results have been found to be relatively consistent for more than 20 years of research, some researchers have proposed that the instructions provided to promote an internal focus were not necessarily appropriate, and as a result, led to the observed detrimental effects [23,24]. In other words, perhaps some internal focusing cues enhance performance while others depress performance. Henceforth, in the present study we sought to test this hypothesis using a variety of task-relevant internal cues within the SLJ, which is a task that has been shown to produce reliable results in previous focus of attention research [12–14,17,20,27–31]. Furthermore, some researchers have questioned the universality of the focus of attention effect, suggesting that an internal focus may in fact promote automaticity in non-experts and that the attentional focus effects may not be generalizable to low or moderately skilled performers [24]. Therefore, the purpose of this study was to test if changing the location of an IFOA resulted in performance changes when executing a SLJ. To achieve this aim, participants were provided different IFOA instructional cues (i.e., toes, knees, hips, arms) compared to a neutral instruction given in a control condition. It was predicted that regardless of the locality of the IFOA instructions, there would be performance detriments to the SLJ when compared to jumps completed in a control condition. The results of the current study confirmed the experimental hypothesis, illustrating all internally focused instructions led to detriments in performance compared to jumps completed in the control condition. These findings are consistent with previous research showing the provision of IFOA instructions inhibits optimal SLJ performance [12,14,17,27–31]. Additionally, our prediction that there would be no differences in jumping performance between the internal conditions was not supported. Specifically, we observed a significant difference between jumps completed in the "knee" condition and the "arm" condition, with further jumps observed in the "arm" compared to the "knee" condition. It is important to note that the jump distance for both of the "arm" and "knee" conditions were significantly shorter compared to the control condition.

The constrained action hypothesis explains how focusing internally leads to decreases in motor performance [2]. According to this hypothesis, an IFOA constrains the motor control systems via conscious control of movements thereby decreasing self-organized automaticity leading to decrements in motor performance. In contrast, adopting an EFOA facilitates self-organized automaticity via unconscious control of the motor system leading to enhanced performance. Findings from the present study show empirical evidence supporting the generalizability of IFOA as hypothesized by Wulf et al. [2]. That is, all internal conditions, regardless of location, resulted in significantly shorter standing long jump distances compared to jumps completed in the control condition. However, the constrained action hypothesis does not explain differences observed between two of the IFOA conditions (i.e., knee & arms). We offer several possible explanations for this finding in the following section.

Previous research using electromyography (EMG), across a variety of motor tasks, corroborate the predictions of the constrained action hypothesis showing increases in muscular activation when using an IFOA with a subsequent performance detriment compared to an EFOA [4,6,34,35]. Specifically, [35] postulated that an IFOA introduced "noise" to the motor control

system that could be the cause for the decrease in performance. Lohse et al. [36] found increased co-contraction of the of agonist and antagonist muscle groups in a simple force reduction task, which demonstrated inefficient muscle recruitment for an IFOA condition when compared to an EFOA. Taken together, we speculate the reason for the significant difference between internal knee and arm conditions could be that within the internal knee condition, there was greater muscular co-contraction of primary jumping muscles (e.g., hamstrings & quadriceps) compared to the internal arm condition during the standing long jump task. A decrease in efficiency of the relevant musculature due to the adoption of an IFOA surrounding the knee joint could explain the performance decrements when compared to decreased efficiency from less relevant musculature found in the internal arm condition.

Another explanation for the difference we observed between arm and knee conditions in the present study could be explained via the distance effect, whereby greater motor performance is achieved by directing attentional resources further away from the body [11]. Previous work from Porter et al. [12] compared external far, external near, and control conditions in a similar SLJ task. Their findings showed that performance was optimized when external cues directed attention further away from the body. Unlike previous work, the present investigation looked at only a variety of internal conditions compared to attempts completed in a control condition. The distance effect may be generalized to include internal cues, whereby, internal cues that direct attention nearest to the center of gravity, result in decreased performance. Under that assumption, one would expect to find distal internal cues focused on the arm swing resulting in greater performance compared to more proximal internal cues focused on the knees. However, we did not see significant differences between a more proximal hip condition relative to the knee. This may be due to the short distance between the center of gravity and the internal cues given for the knee and hip as compared to the larger distance between the center of gravity and the arms. The larger distances may have passed a necessary threshold to elicit a change in performance found in the data observed for this experiment. Hence the results of this study could be the first to provide evidence for the generalizability of the distance effect to internal cues, since the distal arm condition resulted in greater long jump performance compared to the more proximal knee condition.

Momentum is another factor that could contribute to the differences we observed between the knee and arm conditions. Previous research has shown that utilizing arm movement during the standing long jump leads to greater jump distances when compared to restricted or limited arm movement [37–39]. It is possible that a byproduct of instructions directing attention internally to the arms results in an increased generation of upper extremity momentum. This additional momentum, created from the internal arm condition, may have resulted in an increase in long jump distance compared to the knee condition.

The present study sought to investigate if directing attention internally towards different body movements differentially affected the execution of the SLJ. Previous researchers [24,26] have proposed that some internal cues may be enhancing while others are depressing. The results of the present study suggest this point of view may not be accurate in relation to performing the SLJ. That is, jumping performance was depressed relative to a control condition regardless of which form of internal focus was cued. These results confirm predictions made by the constrained action hypothesis and demonstrate that a variety of IFOA instructions reduce standing long jump performance, some worse than others. In conclusion, the findings reported here highlight the importance of avoiding verbal cues which direct attention internally when instructing the SLJ. Furthermore, the findings reported here corroborate previous findings in that jumpers should be cued to direct their attention externally rather than adopting any form of an internal cue.

There are some limitations to the findings of this paper. First, we only had college-aged male novices participate in this experiment. This assisted in preventing the confound of sex-based differences on the SLJ but limits our findings to a specific population. Future work should investigate if a variety of internal focus cues affect motor performance in more diverse populations. Second, our findings are specific to the SLJ; therefore, future work should be conducted on the generalizability of these findings to other motor skills.

## Supporting information

**S1 Data.**
(XLSX)

## Acknowledgments

We would like to thank all the authors for their contributions for this manuscript, and all participants for participating in this study. Additionally, we would like to thank the reviewers which provided thoughtful feedback improving the quality of our paper.

## Author Contributions

**Conceptualization:** Jared M. Porter.

**Data curation:** Andrew J. Strick, Jared M. Porter.

**Project administration:** Jared M. Porter.

**Writing – original draft:** Andrew J. Strick.

**Writing – review & editing:** Andrew J. Strick, Logan T. Markwell, Hubert Makaruk, Jared M. Porter.

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
