## [Decision Letter · Decision Letter 0]

20 Apr 2023

PONE-D-23-04667The location of an internal focus of attention differentially affects motor performancePLOS ONE

Dear Dr. Strick,

Thank you for submitting your manuscript to PLOS ONE. After careful consideration, we feel that it has merit but does not fully meet PLOS ONE’s publication criteria as it currently stands. Therefore, we invite you to submit a revised version of the manuscript that addresses the points raised during the review process.

We look forward to receiving your revised manuscript.

Kind regards,

Monika Błaszczyszyn

Academic Editor

PLOS ONE

2. Please provide additional details regarding ethical approval in the body of your manuscript. In the Methods section, please ensure that you have specified the name of the IRB/ethics committee that approved your study.

Reviewers' comments:

Reviewer's Responses to Questions

**Comments to the Author**

1. Is the manuscript technically sound, and do the data support the conclusions?

Reviewer #1: Partly

Reviewer #2: No

Reviewer #3: Partly

2. Has the statistical analysis been performed appropriately and rigorously? 

Reviewer #1: No

Reviewer #2: No

Reviewer #3: No

3. Have the authors made all data underlying the findings in their manuscript fully available?

Reviewer #1: Yes

Reviewer #2: Yes

Reviewer #3: Yes

4. Is the manuscript presented in an intelligible fashion and written in standard English?

Reviewer #1: Yes

Reviewer #2: Yes

Reviewer #3: Yes

5. Review Comments to the Author

Reviewer #1: The present study investigated the effect of different internal focusing cues in standing long jump performance. The study design was simple, which afforded clear results and interpretations. Also, the methodology was similar to the previous literature, which further strengthens the validity of the study and increases the capacity of the present results to compare and contrast with other studies. However, I have some major concerns regarding the introduction and need clarifications in the statistical analyses.

Introduction

(1) (minor) Line 47 and 48: the ‘motor control system’ and ‘motor behavior’

Are they used synonymously (It seems the latter has a broader meaning)? I recommend using a consistent term if not used purposefully.

(2) (minor) Line 54: ‘…as attention was directed at greater distances from the mover.’

I suggest adding “spatially” or “physically”: e.g., at spatially greater distances. Also, the jump distance increased as the spatial cue of the EFOA moved physically farther, not the jump distance ‘was’ increased, correct?

(3) (minor) The second paragraph: I believe adding more contexts (regarding what parameters or factors have been investigated in SLJ) strengthens the authors’ argument and provides an even better transition to the next paragraph:

(Original) “Since the aforementioned findings, much research using the SLJ has been conducted exploring the distance effect on motor learning and performance with regard to the external far condition [6-10].”

(Recommendation example) I suggest providing specific factors investigated: e.g., “Since the aforementioned findings, various factors of the SLJ have been investigated, including attainability [6,7,10], skill levels [8,9], and xxxxx [REFS]”

(4) (major) The flow of the third and fourth paragraphs is challenging to follow. I recommend a major reorganization of these paragraphs for the following reasons:

a. In the first sentence of the third paragraph, the author brought up three topics: distance, relevance, and breadth of attention. This should be narrowed down.

b. The authors implied, in the first sentence, that the paragraph would be about the distance of IFOA; however, the argument about the distance effect is refuted within the same paragraph, and switched to the topic of relevance without discussing interpretation, providing an approach to consolidate the issue, or providing a theoretical approach to make distinctions between the distance and relevance.

c. The criticism about the relevance was mentioned in the middle of the third paragraph, which was brought up again in the fourth paragraph after discussing different issues. This structure was very difficult to follow.

d. Line 77 – 80 start from: “others have proposed that…” to “the movement techniques” Although these arguments may be reasonable to bring up in the discussion, it seems irrelevant to the present manuscript. I recommend removing these sentences and focusing on the primary topic of the study (i.e., locus of internal foci).

e. The argument of the third and fourth paragraphs is vague: What is the critical factor affecting the previous results introduced here? The distance or relevance? What is the (potential) relationship? What do the authors desire to investigate or clarify in the present manuscript (relevance, distance, or the interaction of both)?

(5) How is the breadth of the attentional focus cues related to the purpose of the present manuscript?

(6) (major) Before the authors provided their hypotheses, it didn’t seem that the problem of the previous studies (distance vs. relevance) was not consolidated:

a. What is the rationale for choosing these different body parts specific to the relevance of the cues and specific to the distance of the cues?

b. What is the rationale for hypothesizing all cues, regardless of the distance and relevance, are equally poorer than an EFOA? For instance, the authors discussed the results regarding the breadth of the focus of attention, which was not different by its factor. However, the authors mentioned that the relevance OR distance did affect the results. Then, the hypothesis is inconsistent with the previous study (Pelleck & Passmore), which is fine, but why?

c. What is the ‘relevance’ of each IFOA cue? Any support from biomechanics, motor control, or strength & conditioning literature?

Method:

• (minor) Participants: Although I am not a strong proponent of blindly relying on the power analysis (because it is more complicated than how they have been used), providing justifications for the sample size is important.

• (minor) Participants: Please justify why only males were recruited (a sentence or two would be sufficient).

• (minor) Participants: How were they recruited? What are the characteristics of the participants? (who can the cohort be generalized?).

• (minor) Apparatus and task: Was the order completely randomized or pseudo-randomized? That is, was there any order effect at all?

Statistical analysis

• (major) How was Type I error controlled?

• (major) Was there any missing data? How did you validate the assumptions? Were there any outliers? How were they decided to be treated before data analysis?

Results:

• (major) How can the degree of freedom be 28 where the sample size was 26?

• Could you provide a table of M and SD?

Figure

• Could you replace the figure with an alternative method? E.g., superimposing raw data over the current bar graph, adding a violin plot next to the current graph, or superimposing the mean and SE points over a box plot (or simply, a box plot). Bar graphs do not show the distribution of the data, so it’s not reader-friendly to demonstrate the validity of the statistical choices.

Reviewer #2: The article entitled “The location of internal focus of attention differentially affects motor performance” presents a study in a long line of research on the effects of attentional focus on performance and learning. Typically, research on attentional focus has centered on the benefits of an external focus compared to a control or internal focus of attention. Consistently, an internal focus of attention has found to be less beneficial than an external focus. The present study sought to investigate various locations of internal focus compared to a control condition. The findings indicate that performance during the internal focus condition was consistently worse on a standing long jump compared to a control condition. Four different locations of internal focus were examined, and a significant difference emerged between performance focus on the knees vs arms.

While the study is generally well written, there are a number of methodological issues that prevent one from drawing conclusions about the results. I will present my general concerns below, followed by specific comments.

General Comments

The authors test a variety of locations of internal focus, but what was the reasoning behind the locations selected? It should be the case that the internal focus locations examined should cover the range of possible variations of internal focus characteristics (e.g., proximal, distal, relevant, irrelevant, etc). The authors should justify their selected choices. For example, if I selected a collections of internal focus locations (e.g,. pointer finger, pinky finger, thumb), I could easily show no difference between internal focus conditions, and potentially wrongly conclude that all internal focus locations are equally detrimental.

The authors claim that an internal focus is detrimental. However, are the authors actually able to claim it is “detrimental”? Does an internal focus actively hurt performance, or does an external focus improve performance relative to internal focus which has no effect on performance at all? For example, if the participants adopted an internal focus, would that result in worse performance, than if they focused on the color blue (assuming that focusing on a color would have zero impact on performance). This leads me to another concern with the methodology. One can’t assume that the control condition acts as a baseline because we don’t know what the participants were focusing on. The lack of manipulation check hampers the ability to determine how to interpret control condition performance. For example, research exists that suggests under control conditions, participants tend to adopt an internal focus of condition. If this happened in the present study, then you essentially have 5 different internal focus conditions, and thus we see differences across internal focus locations in contradiction to the hypothesis of the study.

It appears that the familywise error rate was not taken into account when performing the multiple post hoc comparisons. If one were to implement a Bonferroni correction, then the difference between these two internal conditions would no longer significant. .05/5 = .01 (required significance level).

Specific Comments

Line 46. This is a limited explanation of constrained action hypothesis, and could use a more accurate and detailed explanation of this. For example, explain what is meant when you say system operates more autonomously. Saying that behavior is disrupted doesn’t explain how internal focus hurts performance. This is vague. In what way is it disrupted?

Line 51. You need to better explain what you mean by distance of an EFOA. You need to clarify that you are referring to distal and proximal relationship to the body. A reader with no background in this topic wouldn’t understand what you mean by distance (e.g., distance of what?). You reference this later in the paper, but could be explained earlier when you initially start talking about distance of EFOA.

Line 54. Provide examples of different types of external distance instructions used in this study, in case the reader hasn’t read this article.

Participants: Why only male participants? Were they novice at the SLJ?

Line 121: Please state exactly what the participants were told for the internal focus conditions. What was the exact phrase used?

Line 123: Was the control condition always performed first? No mention of the order of conditions.

Results: Looking at Fig 1, while there are significant differences, it appears that the difference in jump length is only a few centimeters. Is this meaningful?

Line 153: “movers” should be “mover’s”

Line 157: Is it determinantal, or does it just not provide anything helpful?

Line 172: Since you don’t know what people will focus on in the control condition, how can you predict that performance will be different? What if people in the control condition focused on their body movement? Research shows that there is a tendency to adopt an internal focus on sports skills.

Line 178: The difference between these internal conditions appeared to be very small (approximately 2cm). It would be more helpful to provide a relative change in jump length (i.e., 1% increase in jump length). Need to provide the average length of the jumps, then the reader can determine how meaningful 2cm actually is.

Line 186: Can automaticity be anything other than self-organized?

Line 190: “significantly” only with respect to statistics. I’m not convinced this small difference is meaningful.

Discussion: Given the possible reasons for why various internal focus conditions might lead to the observed differences between the legs and arms, why did the researchers predict no differences between the various internal focus conditions. Clearly there are logical reasons why difference might have existed.

Line 215: I’m not sure how arms are considered more distal and legs. From what point are you measuring from?

Reviewer #3: General Comments

A within-participants experimental design was used in this study to investigate the differential effects on horizontal jump distance of adopting four types of internal attentional focus and no attentional focus during standing long jump. It was found that participants jumped farther in the control condition as compared to any of the four internal focus conditions. Participants also had better jump performance when adopting an internal focus on the arm than the knee. There are several major concerns regarding literature review accuracy, methodological rigor, figure accuracy, and basis of discussion points. Several minor recommendations/suggestions were provided in the hope that the authors would find them helpful for improving the overall quality of the manuscript.

Major Comments – Introduction

1. Pg. 4 Lines 72−73: Coker (2018), Neumann and Piercy (2013), Pelleck and Passmore (2017), Schϋcker, Fleddermann, de Lussanet, Elischer, Bohmer, and Zentgraf (2016), Schϋcker, Hagemann, Strauss, and Volker (2009), Schϋcker, Schmeing, and Hagemann (2016), Hill, Schϋcker, Hagemann, and Strauβ (2017), and Oki, Kokubu, and Nagagomi (2018) also compared various types of internal focus cues with external focus cues.

2. Pg. 4 Lines 82−83: See above comment regarding other studies that examined various types of internal focus.

Major Comments – Methods

3. Pg. 5 Line 98: Was this sample size recruited based on a statistical power analysis?

4. Pg. 5 Line 105: As a general guide, only aspects of the task environment that are known to significantly influence motor performance or would be manipulated as required by the design of the intervention need to be described for replication purposes. And for such purposes, adequate details should be provided for reference by other researchers. Does climate control serve these purposes?

5. Pg. 5 Line 109: Specify the thickness of this line.

6. Pg. 5 Lines 112−113: It is mathematically impossible to counterbalance 120 ways of arranging five conditions across 26 participants.

7. Pg. 6 Lines 119−120: Provide examples of how these IFOA cues were used in the instructions given to participants.

8. Pg. 6 Line 131: Were the data checked for normality of distribution to ensure the valid use of ANOVA?

Major Comments – Results

9. Pg. 7 Lines 143−144: Report the direction of this significant difference.

10. Pg. 16 Figure 1: Why are all the error bars of equal length? What are the SE values for these five jump distances?

Major Comments – Discussion

11. Pg. 8 Line 171: The terms within the parentheses are more appropriately described as cues rather than instructions. The use of “in addition” here suggests that internal focus instructions were added to neutral instruction given in the control condition.

12. Pg. 9 Line 189: What does the “generalizability of IFOA” mean?

13. Pg. 10 Line 215−218: Would the hips not be a more proximal IFOA cue than the knees? This “center-of-gravity” hypothesis would predict that an internal focus on the arms would also allow participants to jump farther than an internal focus on the hips. This difference, however, was found to be non-significant in this study.

14. What are the limitations of this study?

15. How could the findings of this study be applied?

Minor Comments

16. Pg. 2 Line 44: Does this emergence refer to the empirical evidence on these differences?

17. Pg. 3 Lines 64−65: Different types of internal focus are not meant to be the same, but may share similar aspects. Consider replacing “equal” with “equivalent”, or its synonym, to reflect this conceptual difference.

18. Pg. 3 Line 68: Use the past tense of “lead” and add “with” after “line”.

19. Pg. 4 Lines 84−85: The conditions per se could not produce the performance, but it is the use of these conditions that could lead to differences in participants’ performance.

20. Pg. 4 Line 86: Consider adding “conditions” after “IFOA”.

21. Pg. 7 Lines 140 and 147−148, and Pg. 9 Lines 180−181: The conditions per se could not make the jump, but the jumps were made by participants under these conditions.

22. Pg. 7 Line 142: Standardize the number of decimal places used to report p values.

23. Pg. 8 Line 166: The use of “argue” is usually associated with the presentation of two opposing views.

24. Pg. 9 Line 175: “led”

25. Pg. 10 Line 205: Add “to” after “due”.

6. PLOS authors have the option to publish the peer review history of their article (what does this mean?). If published, this will include your full peer review and any attached files.

Reviewer #1: No

Reviewer #2: No

Reviewer #3: No

---

## [Author Response · Author response to Decision Letter 0]

31 Jul 2023

We would like to thank the reviewers for there comments. We have attached a word document that includes our responses to each of the reviewers comments.

---

## [Decision Letter · Decision Letter 1]

22 Aug 2023

PONE-D-23-04667R1The location of an internal focus of attention differentially affects motor performancePLOS ONE

Dear Dr. Strick,

Thank you for submitting your manuscript to PLOS ONE. After careful consideration, we feel that it has merit but does not fully meet PLOS ONE’s publication criteria as it currently stands. Therefore, we invite you to submit a revised version of the manuscript that addresses the points raised during the review process.

We look forward to receiving your revised manuscript.

Kind regards,

Monika Błaszczyszyn

Academic Editor

PLOS ONE

Journal Requirements:

Reviewers' comments:

Reviewer's Responses to Questions

**Comments to the Author**

1. If the authors have adequately addressed your comments raised in a previous round of review and you feel that this manuscript is now acceptable for publication, you may indicate that here to bypass the “Comments to the Author” section, enter your conflict of interest statement in the “Confidential to Editor” section, and submit your "Accept" recommendation.

Reviewer #1: All comments have been addressed

Reviewer #3: (No Response)

2. Is the manuscript technically sound, and do the data support the conclusions?

Reviewer #1: Yes

Reviewer #3: Yes

3. Has the statistical analysis been performed appropriately and rigorously? 

Reviewer #1: No

Reviewer #3: Yes

4. Have the authors made all data underlying the findings in their manuscript fully available?

Reviewer #1: No

Reviewer #3: Yes

5. Is the manuscript presented in an intelligible fashion and written in standard English?

Reviewer #1: (No Response)

Reviewer #3: Yes

6. Review Comments to the Author

Reviewer #1: Comments to the authors’ revisions

The revised manuscript has been immensely improved. I agree with most changes. However, there was a slight misunderstanding in response to my comments (Point 6); some of the comments were responded to but without any revisions on the manuscript (point 3, 6, 7, and 8); and one response was unclear (point 5). Most importantly, I cannot agree with the responses to Comments 7, 14, and 15 from the revision (point 2 and 10). Thus, I recommend minor revisions by either clarifying these points and/or providing scientific and rational justifications that go against the current guidelines of science.

(1) Responses to Comment 1 – 6: N/A (resolved)

(2) Response to Comment 7: Just because the previous literature used the same sample size can raise tremendous counterarguments, especially from those who are familiar with methodology/statistics. The lack of justification can lead to statistically unethical procedures, e.g., (a) collecting data until the investigator found significance; (b) stopping data collection when significance was found without justification; (c) simply unreliable results. That is, even if p = 0.05, if the power is low, the conclusion cannot be confidently made about the rejection of the null hypothesis; and (d) none-informative studies, i.e., a small effect size that may not have practical meaningfulness. For example, Porter et al. (2010) collected N = 120, which is one of the largest samples ever collected in the motor learning area. The effect size returned negligible (i.e., d). Other studies collected much less sample size (partially due to the study design), but none of the cited articles in the manuscript (Becker & Smith, 2015; Cocker, 2016, 2018; Ducharme et al., 2016; Hubert & Williams, 2017; King & Porter, 2021; Mirmiran et al., 2019; Nagano et al., 2020; Porter et al., 2010, 2012, 2013, Westphal & Porter, 2012) provided a written justification or reported a priori power analysis. Unfortunately, the research in motor learning is clearly biased and difficult to replicate (McKay et al., 2022a) because of some of the reasons mentioned above. These poor statistical ethics has led to conclusions that many previous research topics in motor learning/control (i.e., autonomy, expectancy, reduced feedback) are highly biased and their effects are negligible (McKay et al., 2022b, 2023), including external/internal focus (Mckay et al., under review).

Given that, just that statement (using the sample size from previous literature with no (report of) prior power analysis) can lead to rejection nowadays. Therefore, for your defense, I strongly encourage the authors to add a limitation that the authors failed to perform a power analysis that can potentially lead to a lack of statistical rigor, and future studies should ensure analysis with high statistical power.

Reference

McKay et al. (under review): https://sportrxiv.org/index.php/server/preprint/view/304

McKay et al. (2022a): https://journals.humankinetics.com/view/journals/jmld/11/1/article-p15.xml

McKay et al. (2022b): https://www.sciencedirect.com/science/article/abs/pii/S1469029222000334

McKay et al. (2023): https://psyarxiv.com/3nhtc/

(3) Response to Comment 8: I was looking for the authors’ response, “Only males were included in the sample to eliminate sex based jumping difference.” I believe this is a sufficient reason to justify the choice of the particular sex. As sex difference and sex inclusion are extremely valued in the current scientific field, please add that response to the manuscript for your defense.

(4) Response to Comment 9: Resolved.

(5) Response to Comment 10: The response does not make sense. If it was truly randomized of 5 potential orders, that’s 120 possible orders? How was the order effect tested? If it was counterbalanced, the order effect could be tested. However, the authors mentioned that ‘no order effects were observed,’ and the term was changed from counterbalanced to randomized. If it was truly randomized, there was no way to test or observe the order effect as it has too many variations, so we just need to hope there would be no bias based on the probability theory. Please explain this in detail.

(6) Response to Comment 11: Type I error is inflation of p-values due to multiple comparisons. I see that the authors performed post hoc tests, not a priori contrasts. (a) How was the inflation of error controlled for the multiple comparisons? (b) Are the p-values of the post hoc tests raw values or values after controlling for the type I error with some methods (e.g., Bonferroni)?

(7) Response to Comment 12(2): Resolved. I meant to ask the authors to add a sentence or the statistical assumption (visual inspection or tests for normality, homogeneity…etc.). The authors have added them in the revised manuscript.

(8) Responses to Comment 12(1,3,4): About outliers. Thank you for the clarification. Please add your responses to the analysis section (i.e., how many outliers and missing data were there; how outliers were treated and the rationale for removing or not removing outliers). As this information affects the future synthesis (if you do not describe missing or outliers, and there was a difference in the degrees of freedom, readers would not be able to tell if it is a typo or due to some data processing), it’s important that the authors add them to the manuscript.

(9) Response to Comment 13: Resolved.

(10) Response to Comments 14 & 15: In response to my recommendations to (a) present M, SD, and (b)replace bar graphs with other forms of figure, the authors responded (a) that would be redundant to figures and (b) bar graphs are consistent with previous literature, and other forms of figures are not more informative than the bar graph.

I cannot agree with both comments. First, the scientific literature should contribute to the development of science. This includes contributions to the future synthesis of knowledge (i.e., meta-analysis) and keeping up with the most recent guidelines based on accumulated knowledge from various areas of science.

For (a), without M, df, and SD (and ideally 95% CI or SE), the literature cannot be used for meta-analysis. This does not only lead to stagnant of scientific development but is also disadvantageous for the authors for not being included in the future synthesis of knowledge. Thus, figures and the actual numbers are not redundant. I see that the authors added M and SE. However, it would be more researcher-friendly if you could add M, SD, and 95% CI to the result section, especially as the authors used SPSS in the analysis. Moreover, the authors added M and SE selectively (I assume only for significant findings). This tradition has been harshly criticized as it produces bias. Thus, either adding all results in the text or simply adding a table would be a better option.

For (b), I clearly see the benefits of the authors’ alternative plots. First, readers can see the spread of the data. Second, readers can see the distribution of the data. Thus, readers can assess whether the results are reliable (i.e., statistical assumptions are met). Third, readers can also see outliers. The debate about how to deal with outliers has not been resolved, and that is why it is important to show and describe the rationale for removing or not removing outliers. Fourth, yes, the alternative graphs may be more visually difficult to see differences; however, the bar graph can lead to exaggerated interpretation as the y-axis range is only 70 – 84 on the left column figures. I applaud the authors for adding asterisks to indicate the source of statistical significance.

There has been a strong shift of moving away from “bad” scientific traditions (not because the past researchers were bad but because there have been improvements), and one of them is the use of bar graphs. Therefore, I respect the authors for trying to be consistent (which is great, and others don’t do that!), but maintaining bad traditions is a different topic. There are numerous criticisms you could find, but for reference, please see Weissgerber et al. (2019).

I do not have a strong position on statistical tests for testing statistical assumptions. There is also a strong shift in statistical consensus that statistical assumptions should be visually inspected. However, there are counterarguments for this. Thus, I do not agree or disagree with the authors’ comment about Mauchly’s test. Regardless, although I applaud the authors for trying different types of figures, I cannot agree with going against scientific development, and any figures on the right column are more informative than the left column figures.

Weissgerber, T. L., Winham, S. J., Heinzen, E. P., Milin-Lazovic, J. S., Garcia-Valencia, O., Bukumiric, Z., Savic, M. D., Garovic, V. D., & Milic, N. M. (2019). Reveal, Don't Conceal: Transforming Data Visualization to Improve Transparency. Circulation, 140(18), 1506–1518.

Reviewer #3: Major Comments – Methods

1. Pg. 8 Lines 173−174: Was a measuring tape used to obtain the exact jump distance from the distal edge of the start line (with respect to the starting position), or the value of the measurement line on the rubber mat that is closest to “the heel of the foot nearest to the start line” used to represent the estimated jump distance?

2. Pg. 8 Lines 178−179: With this correction of the reported order-effect minimization method from counterbalancing to randomization, there is now a methodological concern that amongst the 29 participants, a majority of them might have performed the SLJ with one particular condition as the first of five or the last of five (so that differential effects of warmup, practice, fatigue, etc., might have influenced performance). For each of the five conditions, state the proportion of participants for which it was the first condition as well as that for which it was the fifth (last) condition.

3. Pg. 9 Line 199: It is unclear whether the exception refers to participants not receiving any verbal instructions before each of the two jumps in the control condition, or another unspecified provision to meet condition requirements.

4. Pg. 10 Line 217: The p-value for the Shapiro hip dataset

Major Comments – Results

5. Pg. 10 Line 217: The p-value obtained in the Shapiro-Wilk test result for your hip dataset should be 0.048.

6. Pg. 10 Line 225: Consider reporting the values of standard deviation (SD) alongside these means because SD describes the sample whereas standard error (SD) is not a descriptive statistic.

Major Comments – Discussion

7. The generalizability limitation of this study should be added to the manuscript.

7. PLOS authors have the option to publish the peer review history of their article (what does this mean?). If published, this will include your full peer review and any attached files.

Reviewer #1: No

Reviewer #3: No

---

## [Author Response · Author response to Decision Letter 1]

11 Oct 2023

We appreciate the thoughtful comments from the reviewers as they have helped us refine our manuscript for publication. We have made adjustments to the manuscript in this second round of revisions. We have included more detailed responses to each comment made by the reviewers in our Response to Reviewers document.

---

## [Decision Letter · Decision Letter 2]

30 Oct 2023

The location of an internal focus of attention differentially affects motor performance

PONE-D-23-04667R2

Dear Dr. Strick,

We’re pleased to inform you that your manuscript has been judged scientifically suitable for publication and will be formally accepted for publication once it meets all outstanding technical requirements.

Kind regards,

Monika Błaszczyszyn

Academic Editor

PLOS ONE

Reviewers' comments:

Reviewer's Responses to Questions

**Comments to the Author**

1. If the authors have adequately addressed your comments raised in a previous round of review and you feel that this manuscript is now acceptable for publication, you may indicate that here to bypass the “Comments to the Author” section, enter your conflict of interest statement in the “Confidential to Editor” section, and submit your "Accept" recommendation.

Reviewer #3: All comments have been addressed

2. Is the manuscript technically sound, and do the data support the conclusions?

Reviewer #3: Yes

3. Has the statistical analysis been performed appropriately and rigorously? 

Reviewer #3: Yes

4. Have the authors made all data underlying the findings in their manuscript fully available?

Reviewer #3: Yes

5. Is the manuscript presented in an intelligible fashion and written in standard English?

Reviewer #3: Yes

6. Review Comments to the Author

Reviewer #3: (No Response)

7. PLOS authors have the option to publish the peer review history of their article (what does this mean?). If published, this will include your full peer review and any attached files.

Reviewer #3: No

---

## [Editor Report · Acceptance letter]

3 Nov 2023

PONE-D-23-04667R2 

The location of an internal focus of attention differentially affects motor performance 

Dear Dr. Strick:

I'm pleased to inform you that your manuscript has been deemed suitable for publication in PLOS ONE. Congratulations! Your manuscript is now with our production department. 

Kind regards, 

on behalf of

Dr. Monika Błaszczyszyn 

Academic Editor

PLOS ONE